

**High-resolution automated detection of headwater streambeds for large watersheds**
**Francis Lessard[1,2,3], Naïm Perreault[1,2], Sylvain Jutras[1,2,3]**
[1] Department of Wood and Forest Science, Université Laval, 2405 rue de la Terrasse, G1V
0A6, Québec, QC, Canada
[2] Centre d'étude de la forêt, Université Laval, 2405 rue de la Terrasse, G1V 0A6, Québec,
QC, Canada
[3] CentrEau - Water Research Centre, Université Laval, 1065 avenue de la Médecine, G1V
0A6, Québec, QC, Canada
**Corresponding author:** Francis Lessard, francis.lessard.3@ulaval.ca
**Present address:** Pavillon Abitibi-Price, 2405 rue de la Terrasse, G1V 0A6, Québec, QC,
Canada
**Keywords:** LiDAR, Streambed, Headwater stream, Remote sensing



**Abstract:** Streams are defined by the presence of a streambed, which is a linear
depression where water flows between discernible banks. The upstream
boundary of a stream is called a channel head. Headwater streams, which are
small streams at the top of a watershed, account for the majority of the total
length of streams, yet their exact locations are still not well known. For years,
many algorithms were used to produce hydrographic networks that represent
headwater streams with varying degrees of accuracy. Although digital
elevation models derived from LiDAR have significantly improved headwater
stream detection, the performance of the algorithms with different geomorphic
characteristics remains unclear. Here, we address this issue by testing different
combinations of algorithms using classification trees. Homogeneous
hydrological processes were identified through hydrological classification. The
results showed that in shallow soil that mainly consists of till deposits, the
algorithms that recreate the surface runoff process provide the best explanation
for the presence of a streambed. In contrast, streambeds in thick soil with high
infiltration rates were primarily explained by a small scale incision algorithm.
Furthermore, the use of an iterative process that recreates water diffusion made
it possible to more accurately detect streambeds than other methods tested,
regardless of the hydrological classification. The method developed in this
paper shows the importance of considering hydrological processes when
aiming to identify headwater streams.
words



## 1. Introduction

Streams are characterized by the presence of natural linear depressions, called streambeds. Streambeds, which are mostly formed by fluvial processes, consist of a bed floor and banks, and are identified morphologically. The upstream location of a streambed is generally recognized as being the beginning of a stream and is referred as the channel head. At times, streambeds can be discontinuous or diffuse, leading to subjective identification of streambeds in the field and influence the determined location of the surveyed channel head (Dietrich and Dunne, 1993; Wohl, 2018). On a large scale, headwater streams are extremely important to maintain natural hydrological processes. Indeed, they are representing about two-thirds of the total length of streams in a large watershed (Leopold et al., 1964). Because they have varied ecosystems that include ecotones, headwater streams support rich and diverse fauna and flora (Meyer et al., 2007). In addition, headwater streams provide many ecological services to humans, including good quality drinking water (Alexander et al., 2007; Freeman et al., 2007) and flood control (St-Hilaire et al., 2016). Creed et al. (2017) estimated that for 2.9 million km of headwater streams in the United States, 15.7 trillion US $ in ecological services are provided annually.

Cartographic information on headwater streams at national or provincial scales are largely derived from photointerpretation of stereoscopic aerial photography. This is the main method used for the Géobase du réseau hydrographique du Québec (GRHQ) in Quebec province, Canada. This geodatabase combines and standardizes several sources of hydrographic data, covering an area of 154 million hectares and representing millions of hydrographic features identified from aerial photos. Unfortunately, this method underestimates the true length of streams and is especially inaccurate when identifying





where streams begin and where they become permanent. Streambeds are often
imperceptible on stereoscopic images where only the wide valleys are evident
(Montgomery and Dietrich, 1994).
Other methods based on a digital elevation model (DEM) have been used for several years
to detect streams. These methods, used to produce hydrographic networks, can be divided
into two main categories: channel initiation and valley recognition (Lindsay, 2006). The
channel initiation method can be used to identify the potential locations of streambeds by
thresholding a flow accumulation raster by a minimum drainage area (Band, 1986; Fairfield
and Leymarie, 1991; Jenson and Dominque, 1988; O'Callaghan and Mark, 1984). Valley
recognition can be used to detect streambeds locally through a moving window that
identifies specific pattern depending on the algorithm used (Passalacqua et al., 2012;
Peucker and Douglas, 1975; Tribe, 1992). These methods have been widely used with
coarse resolution DEMs (greater than 10 m) that have generally been derived from aerial
photos.
High resolution geospatial data from LiDAR technology allows for more accurate detection
of headwater streams. These data have recently been made available over large areas,
providing topographic data on the microtopography under the forest canopy and allowing
the creation of DEMs with unprecedented accuracy (Wulder et al., 2008). The
hydrographic networks generated with these new DEMs are much more accurate than those
derived from photointerpretation or those produced from DEMs with a coarser resolution
(Goulden et al., 2014). These DEMs allow for the subdivision of a larger number of small,
previously undetected watersheds, thus generating multiple headwater streams, and
consequently, many branches. Various authors have attempted to use these DEMs to



improve the accuracy of hydrographic networks and the position of channel heads. LiDAR-
derived DEMs have been used to detect streams both locally (Cho et al., 2011; James et
al., 2007) and through channel initiation using a drainage area threshold (Murphy et al.,
2008; Persendt and Gomez, 2016). Other authors have attempted to include the slope to a
flow accumulation raster in order to produce more explicit models (Elmore et al., 2013;
Henkle et al., 2011; James et al., 2010; Montgomery and Foufoula-Georgiou, 1993). While
these methods are more representative of the local impact of water, they still ignore the
heterogeneity of an area and the many other elements that affect bed formation. Among
other things, some authors noted the sensitivity of local flow direction to the elevation error
of the DEM (Hengl et al., 2010; O'Neil and Shortridge, 2013; Schwanghart and Heckmann,
2012). DEMs derived from LiDAR data were also used to quantify the variability of
permanent stream flow lengths, although those studies did not specify where the streambed
begins (Jensen et al., 2018, 2019; Van Meerveld et al., 2019). To the best of our knowledge,
no study has addressed streambed detection using LiDAR data while considering both
channel initiation and valley recognition methods (Heine et al., 2004) on a heterogeneous
territory at the geomorphological level (Wu et al., 2021). Also, no study uses such a large
validation database from real observations acquired in the field.
The main objective of our study is to detect headwater streambeds at a provincial scale.
Our method overcomes the many challenges that have limited this information in the past.
These challenges include highly heterogeneous geomorphological characteristics (such as
surface deposits) and strong anthropization of the land.



**2. Study areas**
The study areas were located in the Appalachian Mountains, St. Lawrence Lowlands,
Southern Laurentides Highlands and Abitibi Lowlands natural provinces, according to the
Quebec Ecological Reference Framework (Fig. 1). This reference framework divides the
territory of Quebec into spatially homogeneous units at various, intertwined levels. The
different levels describe homogeneous units in terms of landform, spatial organization and
hydrographic network configuration (Direction de l'expertise en biodiversité, 2018). The
diversity of the natural provinces thus selected provides a general description of the
headwater streams in Quebec. These natural provinces have distinct hydrological
processes.
The Southern Laurentides Highlands is mostly covered by till, the most widespread surface
deposit in the province of Quebec (Blouin and Berger, 2004; Gosselin, 2002). This natural
province is mountainous, with altitudes varying from 200 to 1200 m. The bedrock mainly
consists of gneiss. Surface deposits are generally thin on summits and steep slopes and
thicker on valley bottoms and gentle slopes. The land in the Southern Laurentides
Highlands is largely forested. In the Appalachian Mountains, the surface deposits are
somewhat similar in distribution to those in the Southern Laurentides Highlands, although
they are thicker in certain areas. However, the bedrock in the Appalachian Mountains is
sedimentary and therefore very different from the Southern Laurentides Highlands. The
altitude here varies from 0 to 1200 m. Unlike the Southern Laurentides Highlands, there is
high anthropization of this natural province due to urbanization and agriculture (Gosselin,
2005a). In the St. Lawrence Lowlands, agricultural activity is also widespread. The surface
deposits in this region are highly heterogeneous and are mainly derived from marine and





glaciolacustrine geomorphic processes. These processes lead to thick soils of sorted
material, including clay and sand. These, in turn, create deposits that range from
impermeable to very permeable. In addition to clay and sand, organic deposits are also
present. The elevation of the St. Lawrence Lowlands is generally less than 100 m, as it was
formed from the Champlain Sea during deglaciation (Gosselin, 2005b). In the Abitibi
Lowlands, the surface deposits are rather thick and consist of silt and clay. These deposits
were produced by marine and lacustrine invasions and are conducive to the formation of
large peatlands. Therefore, the area is relatively flat with altitudes varying from 0 to 350
m. Where present, the bedrock is made of basalt and gneiss (Blouin and Berger, 2002).
Precipitation is not seasonal, but rather constant throughout the year in all study areas.
Precipitation amounts are quite homogeneous and range from 900 mm/year to 1100
mm/year, except in Southern Laurentides Highlands where it can reach 1450 mm/year.
Approximately 20 % of the precipitation falls as snow during the cold season, except in the
coldest regions such as the Abitibi Lowlands and the higher altitude areas of the Southern
Laurentides Highlands where the proportion of snow can reach 30%. Indeed, the average
annual temperature of all the study areas is 3° C to 5° C, except for these two regions where
it is 0° C (MELCC, 2022).



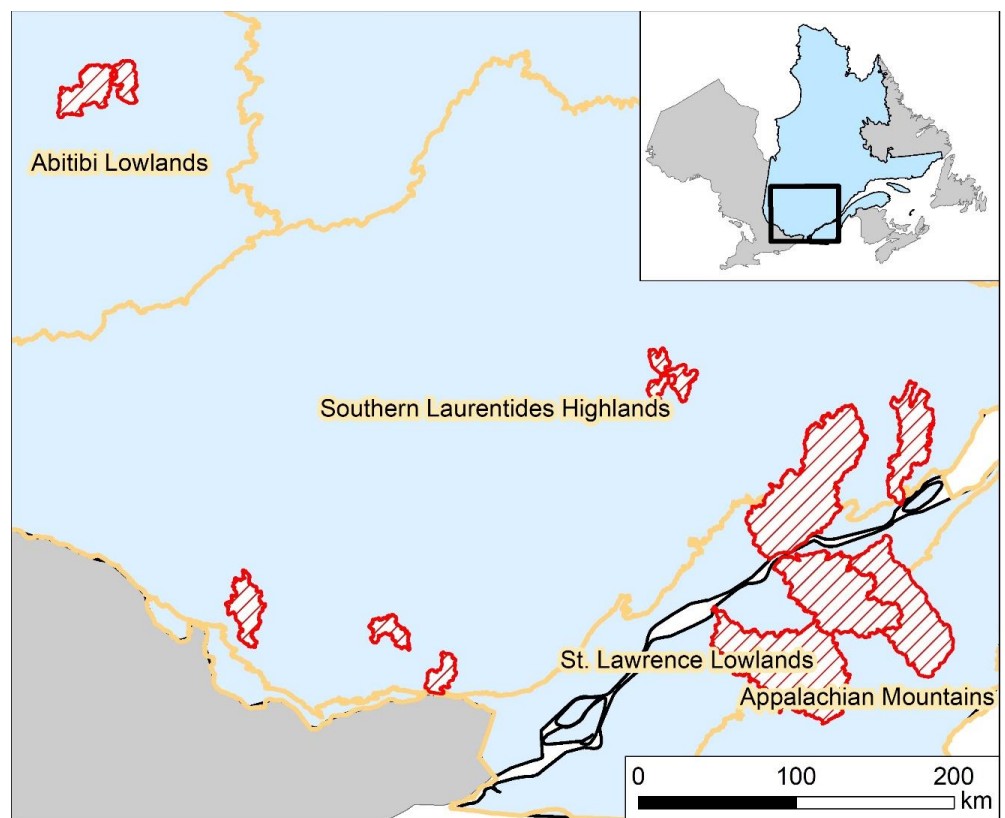

**Figure 1** : Study areas in the Appalachian Mountains, St. Lawrence Lowlands, Southern
Laurentides Highlands and Abitibi Lowlands natural provinces. **[Color is not required
for this figure. Single column fitting figure.]**

**3.      Methods**

*3.1.     Field surveys*

Field based data collection is essential to fully understand stream flow patterns. Field
surveys were conducted from 2017 to 2021 during summer periods using an EOS GNSS
Arrow 100 sub-meter precision GPS. The horizontal accuracy of these devices is
± 0.6 m in open areas and ± 1.2 m in forested areas (Estrada, 2017). These devices were



connected to rugged cell phones in order to use the ArcGIS Field Maps application to
integrate data collection forms as well as relevant background maps.
The positions of streams were recorded from downstream at drainage area generally under
1000 ha to upstream until the streambed completely disappeared. The flow regime, the
width of the streambed, the extent of the water occupation in the streambed and the
presence or the absence of a water flow were collected along de stream path to establish a
high level of understanding. A position was taken on the streams every 50 m or so where
a streambed was present, i.e. where the stream had a bed floor and banks formed by a
fluvial process. Other positions were also taken to identify where there was no streambed.
These information were essential for consistent calibration and validation of streambeds.
To ensure consistent data collection, a 50 m x 50 m grid was used to determine which areas
should be fully surveyed. These areas were mostly located at headwater streams in order
to be able to include channel heads. This procedure was essential to properly assess the
upstream boundary of the headwater streams and precisely record where the streambeds
begin, where they flow from the watershed to the permanent stream, and where they are
absent.
*3.2.    Variables used for analysis*
The geomatic manipulations were mainly performed with the ArcGIS Desktop 10.7
software package, including the Spatial Analyst and 3D Analysis extensions. The open
source SAGA-GIS (Conrad et al., 2015) and WhiteboxTools (Lindsay, 2016a) software's
were also used.
The variables used for analysis were produced from 1 m resolution DEMs of the different
areas. These were generated from LiDAR data from the MFFP (Ministère des Forêts, de la



Faune et des Parcs), with a density of around 2.5 points/m². LiDAR acquisitions were
conducted from 2016 to 2019 (Leboeuf and Pomerleau, 2015), with the exception of a few
areas. The road network was carefully examined in order to include and burn all culverts
that could affect the flow direction (Lessard et al., 2023). Hydrographic networks are
greatly affected by deviations caused by the embankment of the roads. This type of
anthropic influence must therefore be minimized in order to generate coherent flow
direction (Li et al., 2013). Furthermore, the use of a breaching algorithm allowed to
generate hydrologically coherent DEMs prior to hydrographic modeling (Lindsay, 2016b;
Lindsay and Dhun, 2015). Physiographic factors must also be considered during the
modeling process as they significantly influence the location of channel heads and the flow
regime along streams. On the local scale, where the precipitation regime is uniform (Tucker
and Slingerland, 1996), slope, hydraulic force and sediment cohesion generally dictates
streambed formation (Dietrich and Dunne, 1978). The influence of these factors is variable
depending on the type of surface deposit (Dietrich and Dunne, 1993; Dunne and Black,
1970; Montgomery and Dietrich, 1994).
Surface deposits can be used to assess which processes are involved in the formation of a
streambed. Indeed, there are two major types of formation processes. The first type
involves surface processes, which occurs when soil that has low permeability is exposed
to rainfall amounts that exceed the infiltration capacity of the ground, causing surface
runoff (Horton, 1945). Then, when the power of the water exceeds the cohesion of the
sediments, usually in concavities, a streambed forms (Dietrich and Dunne, 1978). The
second type involves subsurface processes that occur when the surface deposits are thick
and infiltrative. Water vertically infiltrates into the ground and eventually reaches



saturation at a junction with the bedrock or an inferior and less infiltrating deposit. Then,
lateral movement of the groundwater occurs. Water emerges from the ground when there
is a change in slope or soil permeability. Streambeds formed in this way tend to be heavily
incised, with flow regimes that are more stable than those formed through surface
processes. Thus, the hydrological response of the streams from subsurface processes is
slightly affected by the intensity of rainfall (Dunne and Black, 1970; Jensen et al., 2019;
Wohl, 2018). Furthermore, it should be noted that there is a gradient between these two
processes for each stream. In order to properly detect streambeds, it is essential to
distinguish these processes through hydrological classification according to surface deposit
type and land use.
Surface deposit mapping has been standardized across the province, including our study
area. Information was collected through photointerpretation conducted several years ago.
Since photointerpretation was mainly used to distinguish forest structures and land use, the
true boundaries of the surface deposits are imprecise, in some cases. Surface deposit
boundaries in agricultural areas are more accurate than those in forested areas because no
other information was mapped during the process. Regardless of these drawbacks,
standardized mapping provides a rough description of the nature and thickness of surface
deposits.
Spatially heterogeneous surface deposits in Quebec have been classified into three
categories and are described in Table 1 (Saucier et al., 1994). The purpose of this
classification step is to differentiate the two types of hydrological processes for headwater
stream formation that were previously described (Dietrich and Dunne, 1993; Lessard,
2020). These classifications consider the infiltration capacity and the water storage





capacity of the ground (Dunne and Black, 1970). The two main variables considered were
the potential thickness and the granulometry of the surface deposits (Dietrich and Dunne,
1993; Wohl, 2018).

**Table 1** : Hydrological classification according to surface deposit types and land use

| Hydrological class | Surface deposits and land use involved |
|---|---|
| Shallow soil | Glacial deposits without morphology such as till, frequent rock outcrops |
| Thick soil with high infiltration rate (including anthropogenic land use) | Glacial deposits with morphology such as moraines, glaciofluvial deposits, fluvial deposits, coarse lacustrine and marine deposits, slope deposits and eolian deposits; Anthropogenic land use were included in this class (Treeless areas including agricultural fields, roads, urbanized areas and powerlines) |
| Thick soil with low infiltration rate | Lacustrine and fine marine deposits, organic deposits |


The first analysis variable, called 'D8', refers to the D8 flow accumulation (O'Callaghan
and Mark, 1984) produced with a 1 m resolution DEM. This variable was selected as it is
the most common algorithm used to produce hydrographic networks. For meaningful
correspondence analysis between this variable and field surveyed streams, the flow
accumulation raster was aggregated at 3 m resolution according to the maximum value.





Then, a maximum focal statistic of two pixels was applied. The purpose of this treatment
was to ensure a 6 m analysis distance between the D8 and the edge of a real stream,
represented in the database by a geospatial line. This prevents the omission error from
being overestimated.
The second analysis variable uses the D8 flow accumulation algorithm while considering
flow direction error due to the elevation uncertainty of the DEM (Hengl et al., 2010;
O'Callaghan and Mark, 1984). This variable, called 'PROB', quantifies the uncertainty
associated with the position of the drainage network. The elevation error in the DEM is
directly related to the uncertainty of the LiDAR data (Wechsler, 2007) and impacts the
position of the hydrographic network (Lindsay, 2006). This type of error is affected by the
landform, and mainly occurs on gentle slopes and slightly convex terrain (Hengl et al.,
2010). Since this type of error is inherent to the shape of the land, it is not affected by the
size of the drainage area implied. The iterative method described in Hengl et al. (2010) was
reproduced in order to create the PROB variable. The method is based on repeatedly
computing a flow accumulation raster from an initial DEM and several altered versions of
the DEM. These altered versions are created by adding random elevation errors to the initial
DEM in order to reproduce the elevation errors from the LiDAR data. The elevation errors
therefore had a standard deviation of 0.08 m, randomly distributed over the DEM. A focal
statistic of 3 m was used on the error raster to ensure the spatial autocorrelation of errors.
Based on the convergence observed by (Lindsay, 2006), 50 iterations were carried out.
Then, each of the flow accumulation rasters were thresholded to a 1.5 ha drainage area to
sum the resulting binary stream network, where a value of 1 indicated the presence of a
streambed and a 0 indicated the absence of a streambed. The matrix of the cumulative value



was then normalized as a percentage to be used as an analysis variable. This PROB variable
revealed the diffusion process of the water in hillsides, where the slope is relatively
uniform. The PROB variable was produced with a 3 m resolution DEM from a 1 m
resolution DEM that was aggregated using the mean values. An average flow accumulation
raster that corresponded to the average of the 50 flow accumulations raster without
thresholding was also produced. This raster was used to create the analysis database and to
calculate the drainage area of the channel heads. To ensure a 6 m analysis distance as well
as the D8 variable, a maximum focal statistic of two cells was performed before summing
or averaging the iterated raster.
The third variable used for analysis is morphometric and allows for the complementary
detection of headwater streams (Lindsay, 2006; Tribe, 1992). The morphometric algorithm
used was the topographic position index, referred to as 'TPI'. This algorithm allowed for
the local detection of small incisions that might represent streambeds (Tribe, 1992). The
scale at which this variable is calculated strongly influences the morphometric feature that
is identified. When the scale is large, the variable will tend to identify valleys, while it
tends towards streambeds when the scale is small (Montgomery and Dietrich, 1992, 1994).
For the purposes of this paper, a relatively small scale of 6 to 30 m was used. This scale is
consistent with the width of the majority of inventoried streambeds. The DEM used to
calculate this variable had a resolution of 2 m and was derived from aggregating a 1 m
resolution DEM with the minimum values. The tool named 'Topographic Position Index'
in SAGA-GIS software was used to produce this variable (Guisan et al., 1999; Weiss,
2001). The TPI variable has not been normalized to keep the homogeneity of the values
between the different study areas.



*3.3.  Analysis database*
In order to perform the subsequent analyses, all actual streambeds were vectorized and geo-
interpreted according to the stream positions recorded in the field. It should be noted that
information on the flow regime was not used in this database. Instead, the presence of a
streambed was used to describe the presence or absence of a stream. Although some beds
have been excavated and channelized, particularly in anthropogenic lands, a bed was
considered to be present when natural fluvial processes allow it to be maintained. The
geospatial lines indicating the exact location of the streambeds were complemented by a
50 m x 50 m grid to represent the complete surveyed area. Thus, areas without a geospatial
line have been assumed to not contain streambeds.
Positions representing the presence of streams were systematically located every 20 m
along geospatial lines that described real streams. Then, positions representing the absence
of a streambed were located according to a sampling principle based on minimum flow
accumulation where it was still possible to observe the presence of a stream. First, within
the grid of the surveyed area, the average flow accumulation raster was thresholded at 0.11
ha. This threshold represents the lowest drainage area of a channel head according to
(Lessard, 2020). Then, the resulting raster was converted to a polygon. Following that step,
a 20 m buffer zone was removed around the geospatial lines that represent real streams.
Finally, absence positions were systematically located according to a hexagonal
distribution in the final resulting polygon. Thus, polygons identifying absence positions
were located only in areas with a minimum 1100 m$^2$ mean drainage area and a minimum
distance of 20 m from any real streams. The number of absence positions was equalized





with the number of presence positions for each natural region within the Quebec ecological
reference framework.
The analysis database was therefore composed of positions describing both the presence
and the absence of streambeds (Fig. 2). The values for the three variables described in the
previous section (D8, PROB and TPI) were extracted for all presence and absence
positions.

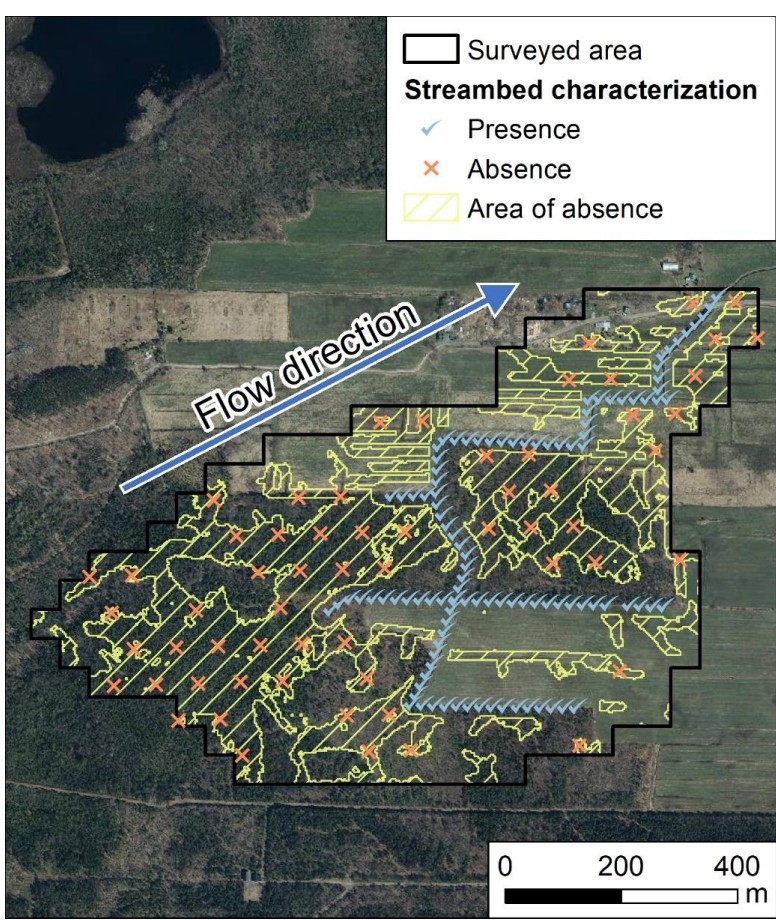


**Figure 2** : Analysis database of positions indicating the presence and absence of
streambeds (Aerial images from continuous imagery of the Government of Quebec;
MRNF). **[Color is not required for this figure. Single column fitting figure.]**




*3.4.    Statistical analysis*

A total of nine logistic regression models were produced, one for each explanatory variable
and hydrologic class combination. Response variable was the presence (1) or the absence
(0) of a streambed. The area under the ROC (Receiver Operating Characteristic) curve was
used to evaluate model performance (Fawcett, 2006). The ROC curve plots the true positive
rate (1 minus omission) relative to the false positive rate (commission). This curve shows
the performance of a given variable by determining the Area Under the Curve (AUC) and
how the increase in the true positive rate will lead to an increase in the false positive rate.
A model with a high AUC will provide a better balance between these two measurements
and will produce better results. Thus, the AUC provides a measure of the ability of the
individual variables to detect a streambed.

Next, four streambed models were compared to each other. Detection performance was
calculated according to hydrological class and using Cohen's kappa, which is a measure of
agreement between the true positive rate and the false positive rate (Cohen, 1960).

The first model examined was the GRHQ. An analysis distance of 6 m was used in order
to compare properly the performance of the GRHQ with the other models. Two of the other
three models corresponded to two different thresholds that were applied to the D8 variable,
which is one of the most commonly used variables for generating stream networks. The
first threshold was the median of the average drainage area of the channel heads surveyed
in the field (referred to as Channel head). The second threshold was the one that maximized
Cohen's kappa for the variable D8 (referred to as Max Kappa). The last model that was
compared is based on a supervised classification approach. This approach groups
observations according to explanatory variables based on previously determined groups,



also known as the response variable. In this case, the response variable was the presence
or absence of a streambed. Classification And Regression Tree (CART) approach was used
because it is simple to apply over a large territory (Breiman et al., 1984). This model was
called CART. One tree was produced for each hydrologic class in order to describe the
formation of headwater streams from homogeneous hydrologic processes. Based on the
literature, different variables were used for each hydrological class. The PROB variable
was the only one that was used to detect streambeds in shallow soil, as the bedrock is
usually close to the surface of the ground and not very suitable for incisions (Jensen et al.,
2018). For the other two hydrological classes in thick soils, the TPI and PROB variables
were used. The surface deposits in these classes are not consolidated, allowing the ground
to be incised. This can then be detected by different morphometric indices (Montgomery
and Dietrich, 1994). The depth and number of branches in the classification trees have been
limited in order to prevent overfitting (Fürnkranz, 1997).

**4. Results**
A total of 464.7 km of streams were surveyed over a known territory of 161.5 km$^2$. The
positions for 1033 channel heads indicating the beginnings of streambeds were determined.
The average drainage areas of the channel head are presented in Fig. 3 using whisker boxes
according to hydrological class. Figure 3 shows that for shallow soil, the average drainage
area is less variable than for thick soils. For thick soils with low infiltration rates, the
average drainage area tends to be higher.



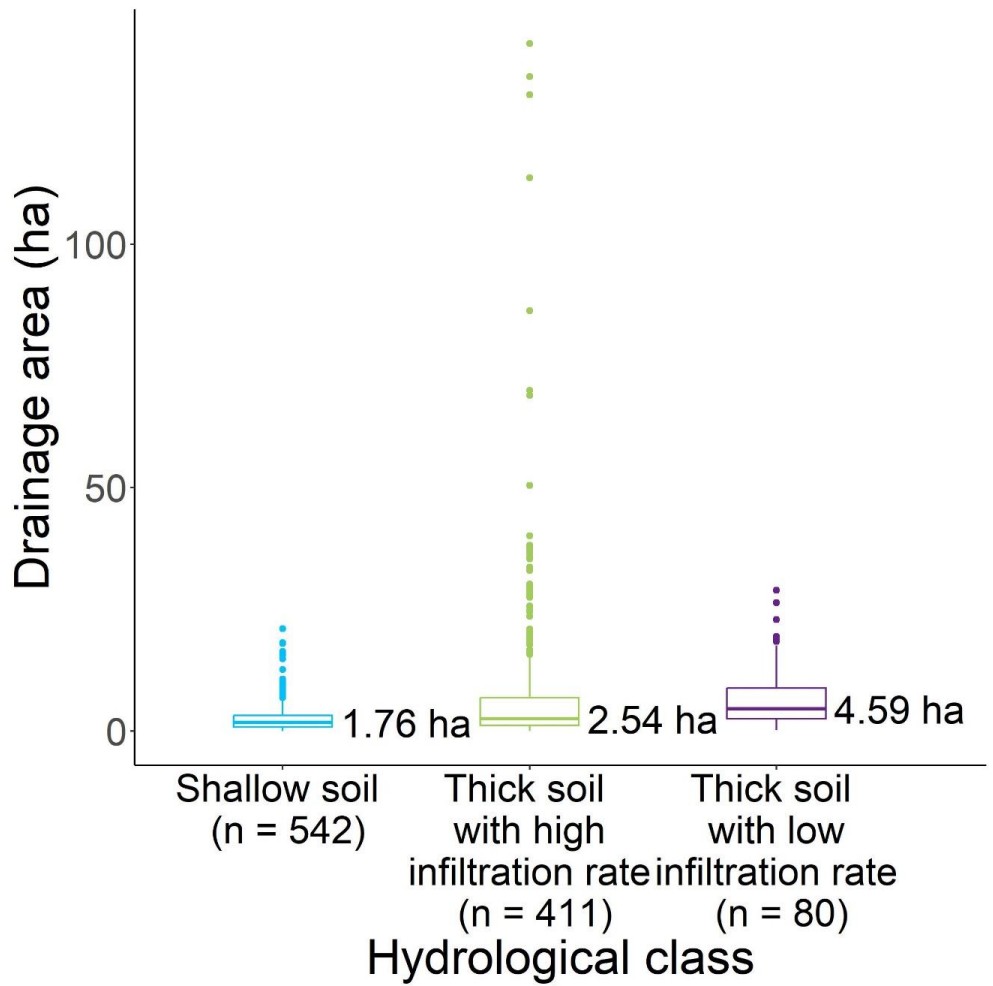

**Figure 3** : Distribution of mean drainage areas of channel heads according to hydrological

class. Median values are shown. **[Color is not required for this figure. Single column**

**fitting figure.]**

The analysis database contains a total of 40 354 positions describing streambeds (20 177

with streambeds present and 20 177 with streambeds absent) located in the entire surveyed

area. A correlation matrix between the analysis variables showed that PROB is negatively





correlated with TPI, with an R of -0.57. This variable therefore identifies where the water
converges, which usually corresponds with the locations of incisions. The other variables
were not correlated with each other.
Three classification trees according to hydrological class are presented in Fig. 4. The tree
for shallow soil shows that when PROB exceeds a threshold of 0.33, a streambed is
generally present. For thick soil with a high infiltration rate, the incision indicated by the
TPI first explains the presence of a streambed. When the incision is greater or equal to -
0.41, indicating a small incision, PROB must be very high in order to indicate the presence
of a streambed, at 0.99. When there is a larger incision, a lower value for PROB can identify
the presence of a streambed. Thus, when the ground is relatively well incised with a TPI
value smaller than -0.41, PROB only needs to be higher than 0.39 to detect a streambed.
In thick soil with a low infiltration rate, PROB provides the initial information regarding
the presence or absence of a streambed. Depending on the different PROB thresholds, TPI
then determines the presence or absence of a streambed.

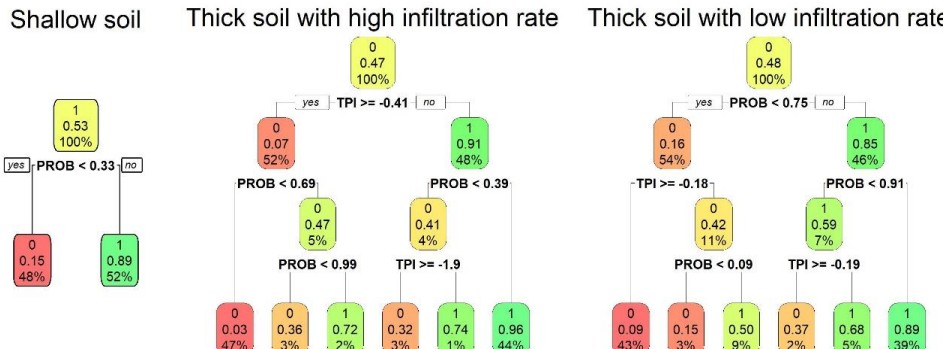


**Figure 4** : Classification trees to detect the presence of streambeds according to variables





D8, PROB and TPI and hydrological class. **[Color is not required for this figure. 2**
**column fitting figure.]**

Figure 5 compares the AUC of individual variables, thus their potential to detect a
streambed. The performance of the four streambed models is also presented. This figure
shows that for the three hydrological classes, PROB performs more effectively than D8
when it comes to detecting streambeds. For thick soil classes, the incision variable TPI has
a higher AUC than D8. For shallow soil, the opposite is true. Compared to the other models,
the GRHQ has a very low true positive rate, meaning it omits many streams regardless of
the hydrologic class. However, the performance of GRHQ is higher for thick soils than for
shallow soils. For shallow soils, although the false positive rate is slightly lower for D8
thresholded with channel heads (Channel head), the Cohen's kappa of the classification
tree (CART) is still higher. The performance of the maximum Kappa of D8 (Max Kappa)
is still very similar to the one of the classification tree (CART). Figure 5 also shows that
the performance of the classification tree (CART) for shallow soil is not in the upper left
part of the ROC curve of the variable PROB. This observation is consistent with the fact
that only this variable was used to calibrate this model. Nevertheless, for both thick soil
classes, the performance of the classification trees (CART) is in the upper left part of the
ROC curve of the variable PROB. This means that the addition of the incision variable TPI
improves the detection of streambeds. For thick soils with high infiltration rates, the two
thresholding methods (Channel head and Max Kappa) yielded similar performances,
although they did not perform as well as the classification tree (CART). The performance
of the classification tree (CART) is also higher than both D8 thresholding methods for thick
soils with low infiltration rates. However, the method using the maximum Kappa (Max



Kappa) yields a higher rate of true positives than the thresholding method using the channel
heads (Channel head).

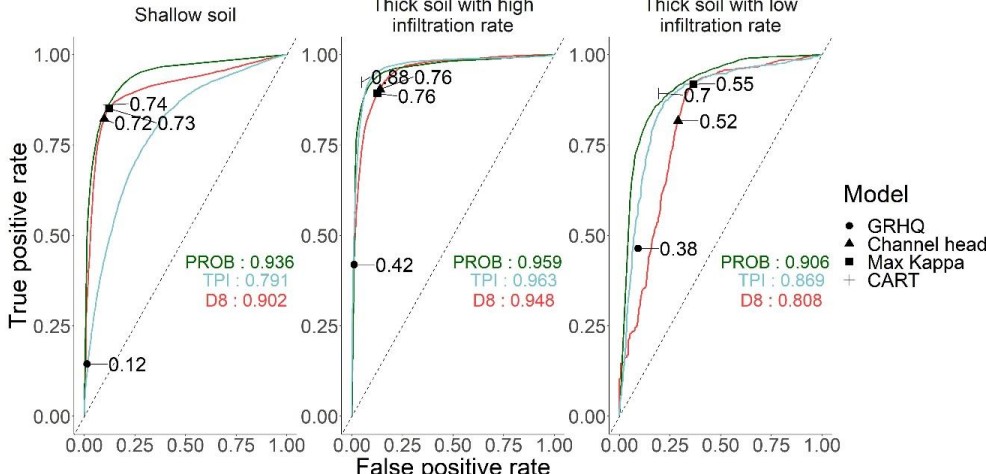


**Figure 5** : ROC curve and AUC values from the logistic regressions of the three variables
according to hydrological class. The performance of the streambed models using Cohen's
kappa is also presented**. [Figure 5 about here. Color is not required for this figure. 2**
**column fitting figure.]**

**5.        Discussion**
The results suggest that the classification tree can detect streambeds more accurately than
the other methods tested. By integrating different topographic indices and ground
information such as surface deposits, the detection of headwater streambeds is much more
efficient in large watersheds, despite the high anthropization of the ground that is
sometimes present. In addition, as the results of the classification trees are rasters (Fig. 6
a)), they can be easily integrated within attribute table of a drainage network by calculating



the mean using a zonal statistic to assess the probability presence of a streambed (Fig. 6
b)). This integration can be done without altering the course or thresholds of the
hydrographic network. Each segment can therefore be truncated according to the presence
or absence of the stream predicted by the model.

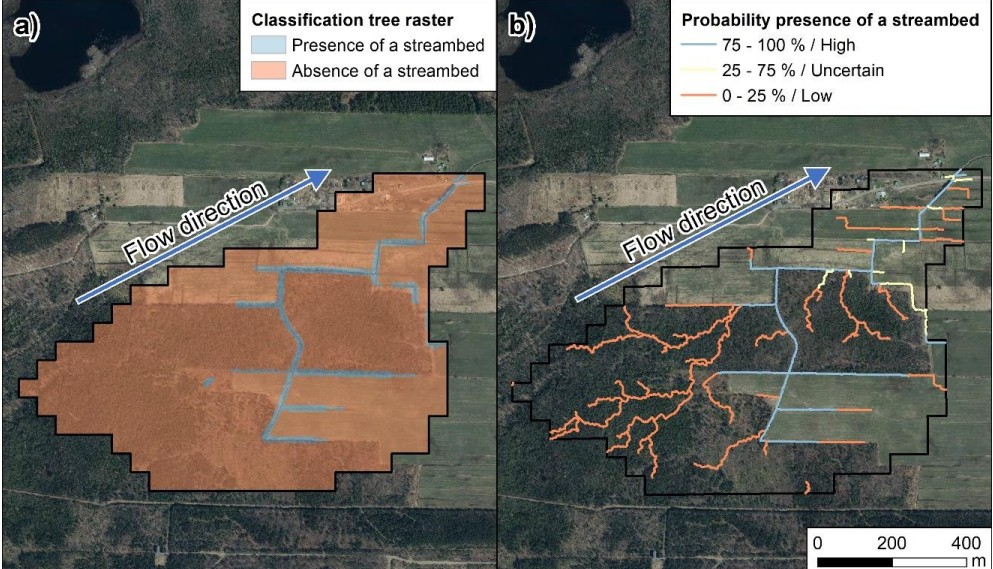



**Figure 6** : Classification tree that has been integrated into the segments of a hydrographic
network to assess the probability presence of a streambed (b) (Aerial images from
continuous imagery of the Government of Quebec; MRNF). **[Color is not required for**
**this figure. 1.5 column fitting figure.]**

The classification tree (CART) drastically increases the true positive rate compared to the
GRHQ. This is because the GRHQ was based on aerial photographs that were primarily
used to characterize vegetation and forest structure. Photointerpretation of these images





did not allow for the detection of streambeds formed by local fluvial processes under the
forest cover (Lessard, 2020). At most, photointerpretation enables the identification of
valleys, for example, on thick soils (Montgomery and Dietrich, 1994). For this reason, the
GRHQ omits fewer streams in thick soil than in shallow soil.
The PROB variable improved the detection of streambeds compared to the conventional
use of only the D8 variable, since it has been thresholded to accurately match the lowest
drainage areas of the channel heads. According to Fig. 3, the 1.5 ha threshold accounts for
the majority of the channel heads. However, the drainage areas of the channel heads are
generally higher for thick soils with low infiltration rates. The majority of the surveyed
streams in this hydrologic class are located in the Abitibi Lowlands natural province. Some
of the drainage areas of the channel heads in shallow soil are smaller than 1.5 ha.
For the shallow soil hydrological class, the PROB variable improves streambed detection
only when a false positive rate of at least 0.12 is specified. Figure 5 shows that for a false
positive rate of 0.25, for example, PROB has a higher true positive rate than the D8
variable. Streambeds that were not omitted with a PROB threshold greater than 0.12 were
mostly small streams with highly variable positions due to the slightly upstream convex
topography (Hengl et al., 2010). It seems that these streambed presence positions have very
low PROB values (48% of these positions have a probability below the 0.33 threshold used;
Fig. 4). The 0.33 PROB threshold enabled a false positive rate that is much lower than
0.25. In fact, the false positive rate was only 0.12. With this 0.33 threshold, the performance
of PROB was almost identical to D8. This is indicated on the figure by the two ROC curves
that were at their closest to each other at approximately the same place as the classification
tree model (CART) (Fig. 5). In order to increase the true positive rate while using the PROB



variable, the threshold could be decreased to allow the smallest streams to be identified.
However, this modification would increase the false positive rate.
The poor performance of the TPI variable for shallow soil is due to the fact that the surface
deposits are generally thin and the slopes are frequently steep. The ground is therefore less
prone to erosion and incision than for the other two hydrological classes (Jensen et al.,
2018; Montgomery and Dietrich, 1994). Indeed, the parameters used to compute TPI do
not enable the detection of small streambeds if they are not located in a valley or in a larger
incision. Furthermore, the hydrological processes involved in this class are mostly surface
flow and not subsurface flow. It is for this reason that D8 and PROB, which tend to be able
to quite precisely recreate surface flow, are the best performing variables in this
hydrological class (Julian et al., 2012; Wohl, 2018).
The incision variable TPI performed better in thick soils with high infiltration rates. This
seems to be due to the fact that unlike shallow soils which are generally thin, infiltrative
soils are thick and unconsolidated. Thus, the main hydrological process for this
hydrological class is a subsurface process, where the water table plays an important role in
the initiation of streambeds. Water infiltrates vertically into the permeable surface deposits
and recharges the groundwater (Dunne and Black, 1970). The locations of the channel
heads do not correspond to specific drainage areas that can be identified by flow
accumulation variables, but rather to local incisions formed by gullying processes where
groundwater intersects the ground surface (Dietrich and Dunne, 1993; Wohl, 2018). This
process occurs where there is a significant change in slope or soil permeability. The
emergence of water from the ground leads to progressive gullying that can be detected by
incision variables (Montgomery and Dietrich, 1994). In this context, groundwater depth

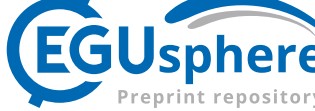

variables such as depth-to-water (DTW; (White et al., 2012)) could be used to explain the
presence of streams in areas where a water table is present. It is important to mention that
the DTW is very sensitive to parameterization and more research is needed for its proper
use (Drolet, 2020).
Streambeds were better detected using solely PROB instead of D8 for thick soils with low
infiltration rates, which occur in territories where there is a high proportion of wetlands
and gentle slopes. The PROB variable mostly reduces the number of commission cases.
For example, in Fig. 5, PROB had a much lower false positive rate than D8 for the same
true positive rate of 0.75. This large reduction in the false positive rate achieved with PROB
reflects the ability of this variable to reproduce a diffuse flow on very flat or slightly convex
terrains (Hengl et al., 2010). Indeed, in 78 % of cases, the positions that correspond to an
absence of a streambed and that are corrected with PROB are wetlands. This is noteworthy
because wetlands represent only 64 % of these positions in this hydrological class. Thus
the PROB variable, using uncertain DEM elevation information, can recreate more realistic
behavior of the water, especially in thick soils with low infiltration rates. By using both
PROB and TPI variables (Fig. 4), streambed detection for this hydrological class can be
improved compared to the use of a single variable. Because the deposits are unconsolidated
and the ground can be incised (Dietrich and Dunne, 1993), the classification tree is in the
upper left part of the ROC curve for the PROB variable as well as for the hydrological class
with the high infiltration. The use of the TPI variable therefore provides an advantage.
A limitation of the classification tree method is that the surface deposit mapping is not
accurate enough for all local hydrological issues. A visual inspection revealed some
inconsistencies in the surface deposit mapping within the same hydrological class.



Another limitation is associated with the anthropization and linearization of natural
streams. While a streambed is the result of a natural fluvial formation process that leads to
ground erosion, an anthropogenic ditch is an artificial bed that is formed by mechanized
digging. However, it is common for naturally formed streambeds to have been excavated
and linearized in agricultural areas. In these cases, it becomes very difficult to distinguish
a streambed from an anthropogenic ditch, even in the field. Excavation concentrates the
flow of water in the artificial bed (Moussa et al., 2002). Thus, an area with previously no
water flow could now be considered a stream (Roelens et al., 2018). Automated detection
methods are therefore likely to be much less reliable in these situations.
We believe that the method described for calibrating the classification tree model is simple
and robust enough to be applied in a different climatic and geomorphic context with local
data describing headwater streambeds. An accurate LiDAR derived headwater streambed
mapping is a powerful tool for government and local organizations involved in water
management and protection.

**6.    Conclusion**
The classification tree method presented in this paper has improved the detection of
headwater streambeds for different hydrological contexts over large watersheds. Reliable
and consistent results were obtained by developing a comprehensive field database. The
variable PROB, which describes the probability of occurrence of a streambed, was used to
correct errors associated with the positioning of streambeds. This variable allowed for
marginal corrections of streambeds in shallow soil, particularly when a high threshold was
used. In order to more precisely explain where streams initiate in shallow soil, variables



characterizing the composition of the upstream watershed such as the average upstream
slope or the composition of deposits should be explored. The variable TPI, which
characterized small-scale incisions, significantly improved the detection of streambeds in
both thick soil hydrological classes when combined with the PROB variable. The small-
scale incision variable worked better in soils with high infiltration rates and the probability
of occurrence worked better in soils with low infiltration rates.
The increased complexity of the methods (inputs and parameterization) makes the
optimizations more difficult for very large territories. It is difficult to integrate the influence
of all physiographic variables into a single model and improvements require multiple
iterations which leads to high complexity. The integration of case studies could improve
models by directly focusing on some of the identified limitations. It is also important to
consider that the input data may sometimes be unreliable, such as those for the road
network, culverts, surface deposits and land use. Thus, developments, such as those
integrating surface deposits, will not be improve if the quality of the raw data remains
unchanged. Visual interpretation of map products and verification by an expert with a good
knowledge of the area is an essential step that should not be neglected under any
circumstances.

**Author contribution**
Francis Lessard and Naïm Perreault contributed to the research project by providing
expertise in methodology, software development, formal analysis, investigation, data
curation, writing, and visualization. Their contributions encompassed various stages, from
data collection and analysis to manuscript preparation.




Sylvain Jutras supervised the project, provided conceptual guidance, and played a role in
writing and reviewing the manuscript. Additionally, Jutras secured funding for the project
and managed administrative tasks related to its execution.

**Competing interest**
The authors declare that they have no conflict of interest.

**Acknowledgements**
The authors thank Quebec's Ministère de l'Environnement et de la Lutte contre les
changements climatiques (MELCC) and Ministère des Forêts, de la Faune et des Parcs
(MFFP), which funded this research project. This project would not have been possible
without the exceptional collaboration of the MELCC's and MFFP's LiDAR mapping
team, together with the many students and research associates who contributed to the
numerous field surveys.

**Data Availability**
Data and code can be found at https://github.com/FraLessard/headwater_streambeds.git,
hosted at GitHub (Lessard and Perreault, 2022).

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
