# Peer review of "High-resolution automated detection of headwater streambeds for large watersheds"

_EGUsphere, 2023_

## Referee Comment (RC1)

**Comments on**

**"High-resolution automated detection of headwater streambeds for large watersheds"**

**Francis Lessard, Naïm Perreault, Sylvain Jutras**

The paper presents different algorithms to automatically detect headwater streambeds from LiDAR DEMs for large watersheds in the province of Quebec, Canada. There is clearly a need to improve the detection of these small headwater streams. Overall, however, I felt that the methodology presented in this study would benefit from being explained in a clearer and more structured way, perhaps with an additional figure to clarify the different steps. The PROB model, in particular, needs to be better described (see detailed comments).

The authors often refer to surface and subsurface processes, with D8 and PROB models representing the surface processes and TPI (topographic position index) the subsurface processes. It is not entirely clear what the role of subsurface processes is in the geomorphological contexts studied here (see detailed comments) and how TPI can account for these subsurface processes, "where the water table plays an important role in the initiation of streambed" (line 477). As stated on p. 14, "The scale at which this variable is calculated strongly influences the morphometric feature that is identified. When the scale is large, the variable will tend to identify valleys, while it tends towards streambeds when the scale is small (Montgomery and Dietrich, 1992, 1994). For the purposes of this paper, a relatively small scale of 6 to 30 m was used."

In summary, I believe the paper presents a useful analysis of headwater streambed detection, but that major revisions are needed to clarify the methodology and make sure the link with hydrological processes is based on more convincing evidence.

***Detailed comments***

p. 2, lines 15-17: I don't think it is necessary to define streambed or channel head in the abstract. I suggest deleting the first two sentences and starting the abstract with "Headwater streams…".

p. 2, line 25: It is not clear what "Homogeneous hydrological processes" means.

p. 2, line 28: Can you really say that the algorithms developed in this study "recreate the surface runoff process", or that "an iterative process" "recreates water diffusion" (line 31)?

p. 3, line 40: I don't understand the "mostly" in this sentence. Aren't streams formed (entirely) by fluvial processes? As with the abstract, I suggest deleting the first two sentences of the Introduction, as the definition of streambeds is well known.

p. 3, lines 56-60: It would be useful to clarify if the underestimation of headwater streams in the GRHQ database is common in other, more widely used, database such as the NHD (National Hydrography Dataset) – see for example Hafen et al. (2020).

p. 4, line 75: Define acronym LiDAR.

p. 4, lines 75-87: This could be summarized in only a few sentences.

p. 5, lines 87-89: Are you still talking about LiDAR DEMs here? Montgomery and Foufoula-Georgiou (1993) worked on a 30-m DEM, not LiDAR, and Henkle et al. (2011) and Elmore et al. (2013) worked from 10-m National Elevation Data DEM.

p. 5, line 91: streambed (instead of bed) formation.

p. 5, line 99: Not entirely clear what the "geomorphological level" means. Also not clear what you mean by "such as large validation database" since you have not mentioned validation database before.

p. 5, line 104: It is the first time you mention anthropization. It would be useful to clarify in the previous paragraphs of the Introduction why this affects the detection of headwater streams (and cite a few references about this).

p. 6, line 112: It is not clear what you mean by "general description" in this context.

p. 6, line 113: It would be useful to give examples of differences in hydrological processes as this sentence is not entirely clear.

p. 6, line 115: It would be useful to provide information on the average slope for hillslopes in the studied watersheds.

p. 8, line 146 (Figure 1): This figure could be improved, adding a legend to clarify what the red polygons represent (watersheds?) and adding a few labels for the main cities (e.g. Montreal), with also a label for the province of Ontario.

p. 9, line 175: software (instead of software's)

p. 10, lines 200-206: References on subsurface processes for streambed formation, and particularly their higher rate of incision, are needed as this explanation is not clear. Subsurface flow will affect the extent of contributing areas (saturated overland flow) and hysteresis (Jensen et al., 2019), but the link between these hydrological processes and streambed incision is not obvious. Channel incision, as noted by Wohl (2018), is more prevalent in arid to semiarid environments, or in karstic landscapes, which is not the geomorphological context of this study.

p. 11, lines 209-211: This is not obvious in the context of this study (see above comment).

p. 12, line 229 (Table 1): It is not clear why anthropogenic land is considered in the same category as glacial moraines and fluvial deposits (thick soil with high infiltration rates). Roads

and urbanized areas, for example, clearly don't have high infiltration rates (nor thick soils). Also, is it always the case that organic deposits have low infiltration rate?

p. 13, line 238: By "geospatial line", you mean a vector line feature?

p. 13, line 244: What is the elevation error in the LiDAR datasets used in this study?

p. 13, lines 253-254: "The elevation errors therefore had a standard deviation of 0.08 m, randomly distributed over the DEM." It is not clear where the value of 0.08 m comes from. Is this based on the LiDAR accuracy or was this value determined by the authors (if so, how?)? Since you conclude that "the variable PROB, which describes the probability of occurrence of a streambed, was used to correct errors associated with the positioning of streambeds" (line 528), it is important for the reader to fully understand this PROB variable which, unlike D8 or TPI, is not a well-known variable. The paragraph starting at line 240 is not sufficiently clear.

p. 14, line 260: "This PROB variable revealed the diffusion process of the water in hillsides, where the slope is relatively uniform." This sentence is not clear and, as mentioned above, it would be useful to provide more information on slope in your study areas (why is slope considered relatively uniform?).

p. 15, line 288: "…a bed was considered to be present when natural fluvial processes allow it to be maintained." Can you clarify what criteria were used to determine when natural fluvial processes allowed to maintain a streambed?

p. 15, line 289: I find the definition of these "geospatial lines" confusing. Were they an output of your analysis? See my previous comment (line 238) where you defined "geospatial lines" as "a real stream" represented in the database.

p. 15, line 298: Lessard (2020) is a Master's thesis (in French). Are there more accessible, peer-reviewed references about this threshold of 0.11 ha?

p. 15, line 302: Confusing units (1100 m$^2$) – this was given in ha on line 298. Overall, it is not straightforward to follow the methodology used in this study to identify headwater streams. Perhaps a summary figure could be provided to clearly indicate the various steps?

p. 18, line 340: "This model was called CART". Delete since this is already mentioned on line 339.

p. 18, line 342: Clarify what you mean by "homogeneous hydrologic processes". Do you mean the 3 hydrological classes described in Table 1 (shallow soil, thick soil with high/low infiltration rates)? There seems to be a confusion in the use of "processes" and "classes", particularly since one of these classes (thick soil with high infiltration rates) includes anthropogenic land where processes are likely different from those in, say, forested watersheds.

p. 18, line 353: Replace "over a known territory of" with "over an area of".

p. 19, line 360 (Figure 3): The font size for X axis labels should be reduced.

p. 19, line 366: Remove "located in the entire surveyed area" as it is implicit.

p. 20, line 384 (Figure 4): Explain the colour scheme in the caption.

p. 21, line 397: Be consistent with upper- or lower-case for kappa.

p. 22, line 422: Were you able to distinguish between highly anthropized areas and more natural areas?

p. 25, line 476-478: What about the urbanized areas that are included in this category (see comments above on Table 1)?

p. 25, lines 481-482: I am still not convinced that you have shown sufficient evidence that in the geomorphological context you are studying, channel heads are formed "by gullying processes where groundwater intersects the ground surface (Dietrich and Dunne, 1993; Wohl, 2018)." The gullies that are described in Dietrich and Dunne (1993) – for example their Fig. 7.24 in northern Tanzania – are really very different from the geomorphological context of this study with thick soils and (often) a vegetation cover. Also, Wohl (2018) stated "Subsurface flow commonly dominates hillslopes with full vegetative cover and thick soils (Dunne and Black, 1970a). Infiltrating water that remains in the subsurface can flow downslope in the unsaturated zone above the water table as throughflow, or in the saturated zone below the water table as ground water. In either case, subsurface water flowing through small interconnected pores will have low velocity, laminar flow (Kampf and Mirus, 2013)."

p. 28, lines 544-546: This sentence needs to be improved (…developments… will not be improve…).

**References**

Hafen, K.C.; Blasch, K.W.; Rea, A.; Sando, R.; Gessler, P.E. (2020) The influence of climate variability on the accuracy of NHD perennial and nonperennial stream classifications. J. Am. Water Resour. Assoc., 56, 903–916

---

## Author Comment (AC1)

**Response to comments on "High-resolution automated detection of headwater streambeds for large watersheds" Francis Lessard, Naïm Perreault, Sylvain Jutras**

The paper presents different algorithms to automatically detect headwater streambeds from LiDAR DEMs for large watersheds in the province of Quebec, Canada. There is clearly a need to improve the detection of these small headwater streams. Overall, however, I felt that the methodology presented in this study would benefit from being explained in a clearer and more structured way, perhaps with an additional figure to clarify the different steps. The PROB model, in particular, needs to be better described (see detailed comments).

The authors often refer to surface and subsurface processes, with D8 and PROB models representing the surface processes and TPI (topographic position index) the subsurface processes. It is not entirely clear what the role of subsurface processes is in the geomorphological contexts studied here (see detailed comments) and how TPI can account for these subsurface processes, "where the water table plays an important role in the initiation of streambed" (line 477). As stated on p. 14, "The scale at which this variable is calculated strongly influences the morphometric feature that is identified. When the scale is large, the variable will tend to identify valleys, while it tends towards streambeds when the scale is small (Montgomery and Dietrich, 1992, 1994). For the purposes of this paper, a relatively small scale of 6 to 30 m was used."

In summary, I believe the paper presents a useful analysis of headwater streambed detection, but that major revisions are needed to clarify the methodology and make sure the link with hydrological processes is based on more convincing evidence.

Thank you for your useful comments. It was indeed difficult to produce a methodology that would allow us to process such a large quantity of data with due regard for the geomorphological context. Your comments will certainly help to clarify our methodology and thereby improve understanding of the results.

*Detailed comments*

p. 2, lines 15-17: I don't think it is necessary to define streambed or channel head in the abstract. I suggest deleting the first two sentences and starting the abstract with "Headwater streams…".

The first two sentences have been deleted.

p. 2, line 25: It is not clear what "Homogeneous hydrological processes" means.

As it's in the abstract, no precision have been made. However, the concept is clarified in the section "3.2. Variable used for analysis" as follow: "The purpose of this classification step is to differentiate the two types of hydrological processes for headwater stream formation that were previously described (Dietrich and Dunne, 1993; Lessard, 2020). These classifications consider the infiltration capacity and the water storage capacity of the ground (Dunne and Black, 1970). The two main variables considered were the potential thickness and the granulometry of the surface deposits (Dietrich and Dunne, 1993; Wohl, 2018)." Furthermore, this sentence have been added : "Thus, the hydrological classes in Table 1 allow us to group together streams whose formation is driven by similar, and therefore theoretically homogeneous, hydrological processes."

p. 2, line 28: Can you really say that the algorithms developed in this study "recreate the surface runoff process", or that "an iterative process" "recreates water diffusion" (line 31)?

The words "recreate" have been replaced by "simulate" as it's a model. Yes, it's well documented that flow accumulation algorithm have the assumption that the ground is an impermeable surface so it can simulate well surface runoff process. For the iterative process, this clarification has been done in the section "3.2. Variable used for analysis" : "This variable allows water diffusion processes to be simulated more adequately than the multiple flow direction algorithms that have been developed for this purpose. Murphy et al., (2009) noted a convergence of results between the single and multiple flow direction algorithms using high-resolution DEMs derived from LiDAR data. The use of a multiple direction algorithm did not provide better results for simulating soil moisture. Indeed, the dendritic flow pattern still appeared visible in the wetlands, even with the use of a multiple flow direction algorithm, probably due to the microtopography present in these DEMs.".

p. 3, line 40: I don't understand the "mostly" in this sentence. Aren't streams formed (entirely) by fluvial processes? As with the abstract, I suggest deleting the first two sentences of the Introduction, as the definition of streambeds is well known.

The term "mostly" have been deleted. The term "mostly" was for other channels that can be formed by gullying process due to gravity (Downslope movement of sediment, Wohl 2018). However, we believe that is important to keep the streambeds definition.

p. 3, lines 56-60: It would be useful to clarify if the underestimation of headwater streams in the GRHQ database is common in other, more widely used, database such as the NHD (National Hydrography Dataset) – see for example Hafen et al. (2020).

Thanks, the reference has been added.

p. 4, line 75: Define acronym LiDAR.

LiDAR acronym is now defined.

p. 4, lines 75-87: This could be summarized in only a few sentences.

This paragraph has been simplified.

p. 5, lines 87-89: Are you still talking about LiDAR DEMs here? Montgomery and FoufoulaGeorgiou (1993) worked on a 30-m DEM, not LiDAR, and Henkle et al. (2011) and Elmore et al. (2013) worked from 10-m National Elevation Data DEM.

Sentence order has been changed in order to properly introduce the use of LiDAR DEMs.

p. 5, line 91: streambed (instead of bed) formation.

The vocabulary has been changed.

p. 5, line 99: Not entirely clear what the "geomorphological level" means. Also not clear what you mean by "such as large validation database" since you have not mentioned validation database before.

The sentence has been modified. The term "validation" has been replaced by "calibration".

p. 5, line 104: It is the first time you mention anthropization. It would be useful to clarify in the previous paragraphs of the Introduction why this affects the detection of headwater streams (and cite a few references about this).

Integration of additional sentences to clarify "anthropisation". References have been added.

p. 6, line 112: It is not clear what you mean by "general description" in this context.
The vocabulary has been changed.

p. 6, line 113: It would be useful to give examples of differences in hydrological processes as this sentence is not entirely clear.
This sentence has been modified.

p. 6, line 115: It would be useful to provide information on the average slope for hillslopes in the studied watersheds.
Indeed, slope information seems important in this context. Average slope will be written or a local slope/drainage area curve will be made for each natural provinces within studied watershed.

p. 8, line 146 (Figure 1): This figure could be improved, adding a legend to clarify what the red polygons represent (watersheds?) and adding a few labels for the main cities (e.g. Montreal), with also a label for the province of Ontario.
This sentence have been added : "Red polygons represent watersheds where field surveys were carried out." This figure has been improved.

p. 9, line 175: software (instead of software's)
This sentence has been modified.

p. 10, lines 200-206: References on subsurface processes for streambed formation, and particularly their higher rate of incision, are needed as this explanation is not clear. Subsurface flow will affect the extent of contributing areas (saturated overland flow) and hysteresis (Jensen et al., 2019), but the link between these hydrological processes and streambed incision is not obvious. Channel incision, as noted by Wohl (2018), is more prevalent in arid to semiarid environments, or in karstic landscapes, which is not the geomorphological context of this study.

p. 11, lines 209-211: This is not obvious in the context of this study (see above comment).
The incision perceptible in our study area are made through longer process occurring since beginning of Holocene and are under canopy unlike in arid to semiarid environments. As noted in figure 3, drainage area are way more variable in the hydrological class "Thick soil with high infiltration rate" that other classes. This is due to the contribution of subsurface process for the creation of streambeds. Indeed, in this class, a certain number of stream are formed by water coming out of the ground, i.e. resurgences (larger number compared to other hydrological classes). A figure will be added in order to show the contribution of incision and TPI for this class. Furthermore, the figure 6 will me modified. The following territory will be used instead:

[Figure]

[Figure]

p. 12, line 229 (Table 1): It is not clear why anthropogenic land is considered in the same category as glacial moraines and fluvial deposits (thick soil with high infiltration rates). Roads and urbanized areas, for example, clearly don't have high infiltration rates (nor thick soils). Also, is it always the case that organic deposits have low infiltration rate?
This sentence has been modified. Roads and urbanized areas are not represented in the database. Yes, organic deposits always have a low infiltration rate when saturated and then contribute to streambed formation through the surface runoff process. A reference has been added to clarify this point.

p. 13, line 238: By "geospatial line", you mean a vector line feature?
This sentence has been modified.

p. 13, line 244: What is the elevation error in the LiDAR datasets used in this study?
This is a general statement.

p. 13, lines 253-254: "The elevation errors therefore had a standard deviation of 0.08 m, randomly distributed over the DEM." It is not clear where the value of 0.08 m comes from. Is this based on the LiDAR accuracy or was this value determined by the authors (if so, how?)? Since you conclude that "the variable PROB, which describes the probability of occurrence of a streambed, was used to correct errors associated with the positioning of streambeds" (line 528), it is important for the reader to fully understand this PROB variable which, unlike D8 or TPI, is not a well-known variable. The paragraph starting at line 240 is not sufficiently clear.
A reference was added. A new figure has been added to illustrate the variable PROB.

p. 14, line 260: "This PROB variable revealed the diffusion process of the water in hillsides, where the slope is relatively uniform." This sentence is not clear and, as mentioned above, it would be useful to provide more information on slope in your study areas (why is slope considered relatively uniform?).
This sentence has been modified.

p. 15, line 288: "…a bed was considered to be present when natural fluvial processes allow it to be maintained." Can you clarify what criteria were used to determine when natural fluvial processes allowed to maintain a streambed?
This sentence has been modified.

p. 15, line 289: I find the definition of these "geospatial lines" confusing. Were they an output of your analysis? See my previous comment (line 238) where you defined "geospatial lines" as "a real stream" represented in the database.
The vocabulary has been changed. The formulation of the sentences in the context seems effective.

p. 15, line 298: Lessard (2020) is a Master's thesis (in French). Are there more accessible, peer reviewed references about this threshold of 0.11 ha?
No, there are no more accessible and peer-reviewed references on this threshold.

p. 15, line 302: Confusing units (1100 m2 ) – this was given in ha on line 298. Overall, it is not straightforward to follow the methodology used in this study to identify headwater streams. Perhaps a summary figure could be provided to clearly indicate the various steps?

This sentence has been modified.

 "This model was called CART". Delete since this is already mentioned on line 339. This sentence has been deleted.

 Clarify what you mean by "homogeneous hydrologic processes". Do you mean the 3 hydrological classes described in Table 1 (shallow soil, thick soil with high/low infiltration rates)? There seems to be a confusion in the use of "processes" and "classes", particularly since one of these classes (thick soil with high infiltration rates) includes anthropogenic land where processes are likely different from those in, say, forested watersheds.

See comment above about homogeneous hydrological processes and anthropogenic land uses. The term "processes" refer to the 2 main categories of surface and subsurface flow and the term "classes" refer to the classification use in this article.

 Replace "over a known territory of" with "over an area of". This sentence has been modified.

 The font size for X axis labels should be reduced. The front size for X axis has been reduced.

 Remove "located in the entire surveyed area" as it is implicit. This sentence has been modified.

 Explain the colour scheme in the caption. The colors red, orange, yellow and green illustrate very low, low, medium and high probability respectively.

 Be consistent with upper- or lower-case for kappa. The performance is measured with Kappa and can therefore be confusion as one of the model is created with the max Kappa of only D8 variable used alone (that's why the model is on the line of the D8 ROC curve in figure 5). See line 333-334 for details about how the model was created.

 Were you able to distinguish between highly anthropized areas and more natural areas?  What about the urbanized areas that are included in this category (see comments above on Table 1)? This sentence has been modified. High anthropization as roads and urbanized areas are not represented in the database.

 I am still not convinced that you have shown sufficient evidence that in the geomorphological context you are studying, channel heads are formed "by gullying processes where groundwater intersects the ground surface (Dietrich and Dunne, 1993; Wohl, 2018)." The gullies that are described in Dietrich and Dunne (1993) – for example their Fig. 7.24 in northern Tanzania – are really very different from the geomorphological context of this study with thick soils and (often) a vegetation cover. Also, Wohl (2018) stated "Subsurface flow commonly dominates hillslopes with full vegetative cover and thick soils (Dunne and Black, 1970a). Infiltrating water that remains in the subsurface can flow downslope in the unsaturated zone above the water table as throughflow, or in the saturated zone below the water table as ground water. In either case, subsurface water flowing through small interconnected pores will have low velocity, laminar flow (Kampf and Mirus, 2013)."

See comment above about incision with the new figures.

p. 28, lines 544-546: This sentence needs to be improved (...developments... will not be improve...). This sentence has been modified.

References Hafen, K.C.; Blasch, K.W.; Rea, A.; Sando, R.; Gessler, P.E. (2020) The influence of climate variability on the accuracy of NHD perennial and nonperennial stream classifications. J. Am. Water Resour. Assoc., 56, 903–916

---

## Author Comment (AC2)

**Response to comments # RC2 on "High-resolution automated detection of headwater streambeds for large watersheds" Francis Lessard, Naïm Perreault, Sylvain Jutras**

Dear Authors,

The developed approach is interesting and might be applicable for different landform and climate contexts. The presented work is also a good basis for further studies, which may consider streamflow regimes and shallow groundwater processes to detect headwater streambeds. However, I think the manuscript must be improved prior to its publication, especially regarding to the presentation of results. Please, see below my suggestions and comments:

1. Describe in detail the specific objectives of the study.

**More details about specific objectives have been added. This will give a better understanding of the method and why it's a novelty.**

2. You should provide some photographs highlighting the main characteristics of the study area as supplementary material.

**Field photographs have been added as supplementary material to show the gradient of stream types according to hydrological process.**

3. In the text, you mention several times the word "permanent" relating to "stream". However, I think you mean "perennial".

**The term have been modified to be consistent with the literature.**

4. Table 1: I do not think that roads and urbanized areas have high infiltration rates.

**In fact, none of the streams surveyed in the field are located on roads or in urbanized areas. It has been modified to focus on Quaternary deposits rather than land use.**

5. Please, provide a flowchart with the methodological steps of the work in the beginning of the methodological section. A short introduction of the applied approach is also valuable.

**A simple flowchart has been added in the beginning of the methodological section with a short introduction to ensure ease of understanding.**

6. Figure 3: show y-axis in logarithmic scale.

**Since the figure represents channels head, the range is limited, and the logarithmic scale is not the best way to show drainage area. However, the limits of the y-axis have been modified to provide a clearer visualization of the data.**

7. You found that PROB is negatively correlated with TPI, with an R of -0.57. Does this multicollinearity have no impact on the presented classification tree models in Fig. 4?

**No, multicollinearity is not an issue in classification trees because they make binary decisions based on individual variables, independently selecting the most informative variables for splitting at each node.**

8. Please, improve the presentation of your results, giving more details about them. Moreover, what else can be explored or assessed from the surveyed data? Are there any spatial patterns? What if you compare the results from the different natural provinces?

**We went into more detail on how to interpret the results and try to make connections according to spatial patterns. Unfortunately, it will not be possible to compare the results with other natural provinces, as we do not have field data to confirm these results.**